# Land use driven change in soil pH affects microbial carbon cycling processes

Ashish A. Malik [1,8], Jeremy Puissant[1], Kate M. Buckeridge[2], Tim Goodall[1], Nico Jehmlich [3], Somak Chowdhury[4], Hyun Soon Gweon [1,9], Jodey M. Peyton[1], Kelly E. Mason [5], Maaike van Agtmaal[6], Aimeric Blaud[7], Ian M. Clark [7], Jeanette Whitaker [5], Richard F. Pywell[1], Nick Ostle[2], Gerd Gleixner [4] & Robert I. Griffiths [1]

Soil microorganisms act as gatekeepers for soil–atmosphere carbon exchange by balancing the accumulation and release of soil organic matter. However, poor understanding of the mechanisms responsible hinders the development of effective land management strategies to enhance soil carbon storage. Here we empirically test the link between microbial ecophysiological traits and topsoil carbon content across geographically distributed soils and land use contrasts. We discovered distinct pH controls on microbial mechanisms of carbon accumulation. Land use intensification in low-pH soils that increased the pH above a threshold (~6.2) leads to carbon loss through increased decomposition, following alleviation of acid retardation of microbial growth. However, loss of carbon with intensification in near-neutral pH soils was linked to decreased microbial biomass and reduced growth efficiency that was, in turn, related to trade-offs with stress alleviation and resource acquisition. Thus, less-intensive management practices in near-neutral pH soils have more potential for carbon storage through increased microbial growth efficiency, whereas in acidic soils, microbial growth is a bigger constraint on decomposition rates.

[1] Centre for Ecology and Hydrology, Wallingford OX10 8BB, UK. [2] Lancaster Environment Centre, Lancaster University, Lancaster LA1 4YQ, UK. [3] Department of Molecular Systems Biology, Helmholtz Centre for Environmental Research-UFZ, Leipzig 04318, Germany. [4] Department of Biogeochemical Processes, Max Planck Institute for Biogeochemistry, Jena 07745, Germany. [5] Centre for Ecology and Hydrology, Lancaster LA1 4AP, UK. [6] Imperial College London, Ascot SL5 7PY, UK. [7] Department of Sustainable Agriculture Sciences, Rothamsted Research, Harpenden AL5 2JQ, UK. [8] Present address: Department of Ecology and Evolutionary Biology, University of California, Irvine 92697, USA. [9] Present address: School of Biological Sciences, University of Reading, Reading RG6 6UR, UK. Correspondence and requests for materials should be addressed to A.A.M. (email: a.malik@uci.edu)

The need for food and energy for the growing human population has led to an immense pressure on the planet's soil resources, with intensive land management often leading to reduced soil organic carbon (SOC) storage[1,2]. Land management strategies to reverse this decline are needed, but we do not sufficiently understand the detailed mechanisms underlying the accumulation and loss of SOC. This is hampering the development of better land management strategies and introduces uncertainties into models that forecast feedbacks to climatic change. Furthermore, soil is the largest carbon (C) pool in the terrestrial biosphere containing twice as much C as in the atmosphere, with the topsoil (0–30 cm) containing approximately half the amount[3–5]. By regulating the storage and release of organic C through decomposition of soil and plant organic matter, soil microorganisms play a major role in soil C fluxes[4,6]. New research suggests that better knowledge of this microbial regulation may be critical in projecting changes in SOC, thereby improving predictions of climate change feedbacks[7,8]. A new paradigm also recognizes the direct, significant contribution of microbial biomass to organic matter accumulation, whereby microbial residues are transformed into stable SOC fractions[4,9–11]. However, uncertainties remain over the influence of microbial physiology on its biomass production and eventual persistence in soil. This highlights the need to explore both the ecological and biochemical basis for SOC storage with a focus on the microbial traits—their phenotypic characteristics[12,13].

Shifts in microbial traits due to climate change have been shown to have consequences for SOC storage[14,15]. Microbial carbon use efficiency (CUE) or growth efficiency, the proportion of substrate C that microorganisms assimilate versus that lost in respiration, is a key trait that determines the fate of C in soils[14,16,17]. Recent theory suggests that high microbial growth efficiency may indicate increased ability of those communities to store SOC through relatively greater biomass synthesis and resultant increases in the amount of microbial residues available for stabilization[11,14,15,18,19]. An increased microbial investment in resource acquisition in the form of extracellular enzyme production to degrade complex substrates is thought to result in lower growth efficiency[20]. Similarly, physiological adaptations to alleviate stresses such as soil acidity or decreased water availability could lead to increased maintenance energy requirements, thus lowering the growth efficiency[21]. Yet, microbial trait trade-offs between the use and acquisition of complex resources, adaptation to stress and biomass production have hitherto been unexplored. We argue that these trade-offs are significant in determining the fate of C inputs in soils.

The collective traits of diverse microbial populations regulate the ecosystem functioning through their interactions with each other and the environment[12]. We now have evidence that gradients of soil properties such as soil pH are strong drivers of soil microbial diversity[22,23]. Therefore, we posit that along similar gradients, there are differences in microbial ecophysiological traits that may affect CUE at a community level. Knowledge of such empirical trends could be the key to better understand the effects of land management on SOC storage. Identifying consistent effects of land management on soil microbial ecophysiology is difficult as land use intensification and their impacts on edaphic properties like soil pH, bulk density and moisture content are often site-specific and context-dependent[15,24]. Trait estimates from literature syntheses provide the theoretical and empirical basis of the CUE variability across soils and suggest that microbial efficiency is an integrative measure of stoichiometric constraints, substrate quality and quantity, soil biodiversity, and edaphic properties[25–27]. However, we still cannot confidently establish empirical relationships between the microbial CUE and environmental variables. This makes it difficult to predict the

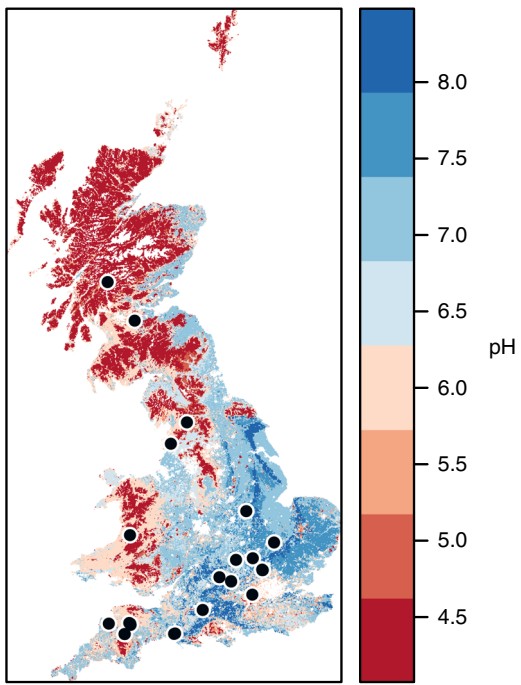

**Fig. 1** Geographical distribution of sampling sites. Soil sampling locations across Britain are displayed over a soil pH map of Britain created using maptools [https://CRAN.R-project.org/package=maptools] and gstat [https://CRAN.R-project.org/package=gstat] packages under the R environment software; pH data was derived from the UK Soils portal (ukso. org). Soils were sampled from 56 sites, and 21 local land use contrasts were available to study the effects of land use intensification. Symbols of sites in close proximity overlap in the map

spatiotemporal patterns of CUE that are key to upscaling the microbial ecophysiological responses to anthropogenic change[28].

The overall aim of this study was to assess the effect of land use intensity on microbial ecophysiological traits across landscape soil gradients and evaluate the consequences of trait trade-offs on SOC accumulation. To achieve this, we analyzed the microbial traits in soils collected from 56 distributed sites across the UK (Fig. 1) in temperate habitats ranging in land use from low-intensity species-rich grasslands to high-intensity grasslands and croplands. We used regression and path analyses to discern potential microbial ecophysiological controls on SOC accumulation and associated environmental drivers. Using stable carbon isotope tracing and soil metaproteomics, we demonstrate two distinct mechanisms of SOC accumulation at low and high soil pH. The mechanisms highlight the significance of microbial growth and metabolic efficiency on SOC accumulation. This collective knowledge enables the changes in soil pH, induced by land use intensification, to be used as a proxy to determine the effect of land management strategies on microbial soil carbon cycling processes.

## Results

**Empirical links between microbial physiology and soil carbon.** We first tested the broader relationships between microbial ecophysiology and soil properties, using C concentration as a proxy for SOC accumulation, to understand how microbial processes determine the mineralization, assimilation, and accumulation of organic matter inputs which underpin SOC formation and loss. For soils sampled from a landscape-scale gradient of 56 spatially dispersed sites across Britain (three replicates per site), we observed clear relationships between certain microbial

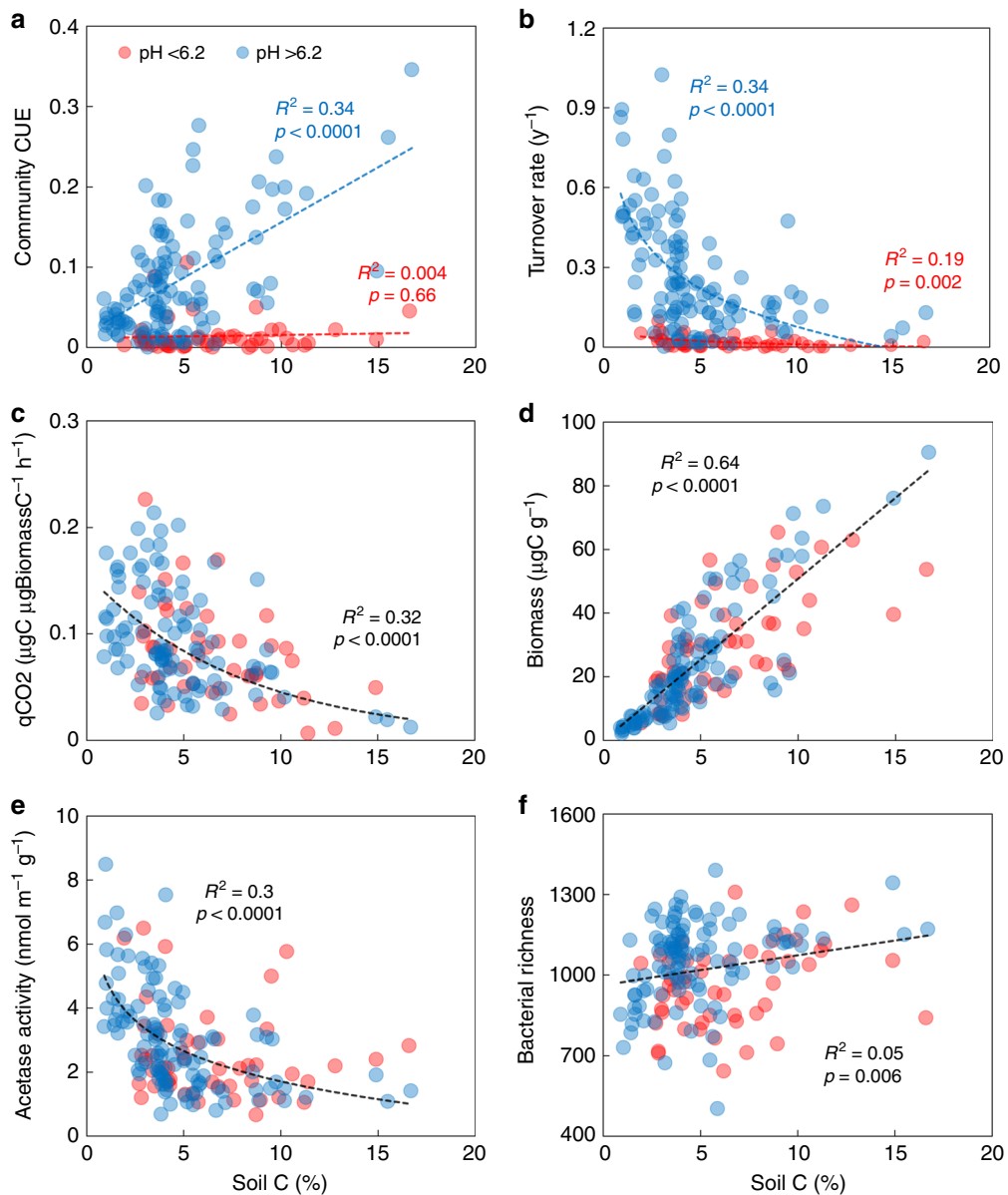

**Fig. 2** Relationship between microbial parameters and soil carbon. Regression trends of microbial community CUE–carbon use efficiency (**a**), turnover or growth rate (**b**), biomass-specific respiration or qCO2 (**c**), DNA-C concentrations as biomass proxy (**d**), extracellular enzyme investment (**e**), and bacterial taxonomic richness (**f**) with SOC concentrations across the landscape-scale gradient of soils. Data from all 56 sites with three replicates at each site are presented here as independent points (red circles: pH < 6.2; blue circles: pH > 6.2). In **c–f**, there were no partitioning of microbial traits across the threshold pH value of 6.2, and the black regression lines include all data points. Best-fitting regression models were: **a** pH < 6.2: $n = 50$, $y = 0.0004x + 0.01$ and pH > 6.2: $n = 113$, $y = 0.01x + 0.02$; **b** pH < 6.2: $y = -0.02 \ln(x) + 0.05$ and pH > 6.2: $y = -0.21 \ln(x) + 0.55$; **c** all data: $n = 163$, $y = 0.16e^{-0.12x}$; **d** all data: $y = 5.09x - 0.07$; **e** all data: $y = -1.37 \ln(x) + 4.87$; **f** all data: $y = 11.06x + 962$

ecophysiological measures (biomass estimates and specific respiration) and SOC concentration. However, substrate CUE at a community level, measured as the proportion of $^{13}$C-labeled plant litter DOC incorporated into microbial DNA relative to that lost as respired $CO_2$, appeared to be driven by interactions of multiple edaphic properties and could not be assembled in a single linear model. We then used recursive partitioning through regression tree analysis to disentangle the interactions; the best partitioning parameter was soil pH from among the edaphic properties that were tested, which also included soil moisture, clay content, C and N concentration, and C:N ratio (Supplementary Fig. 1). The threshold was determined at pH 6.2 using slope failure test (piecewise regression) for CUE versus SOC linear regression (Supplementary Fig. 2)[29]. In other words, community CUE

measured here was distinctly different across the pH threshold of 6.2; above this value community CUE and SOC concentration co-varied, indicating the interdependence of microbial growth efficiency and SOC accumulation (Fig. 2a). The CUE–SOC relationship broke down below the threshold pH, where microbial growth constraints imposed by acidity and wetness could be more important in organic matter accumulation. In such acidic soils, microbial turnover or growth rate (amount of new DNA formed per unit time) was very low (Fig. 2b), but the respiratory response did not differ across the entire range of soil pH (Fig. 2c); both in line with previous studies[30–32]. It is plausible that in lower pH soils, there is a shift from growth to maintenance respiration as a trade-off to increased investment into physiological strategies to survive in a stressful acidic environment that can often also be

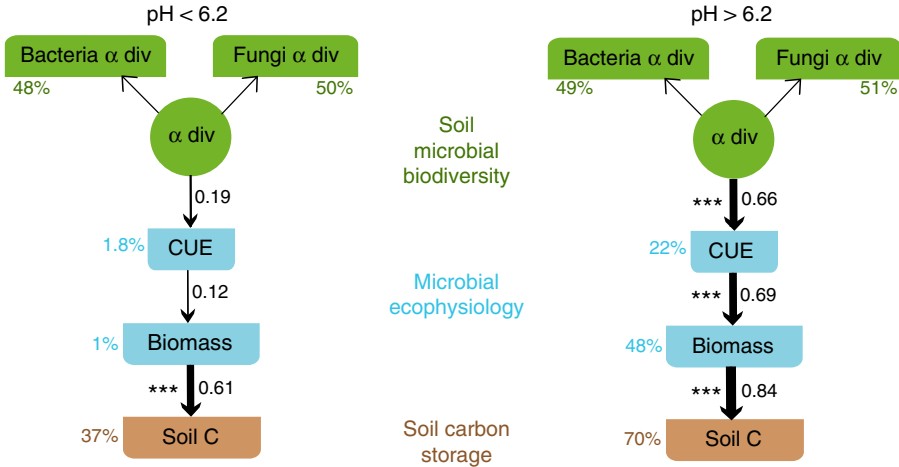

**Fig. 3** Model outcome of expected causal relationships. Most fitting paths of causality obtained through structural equation modeling for the two datasets across the soil pH threshold value of 6.2 (pH < 6.2: $n = 50$, CFI = 1, RMSEA = 0, SRMR = 0.079, AIC = 693, $P = 0.67$; pH > 6.2: $n = 113$, CFI = 1, RMSEA = 0, SRMR = 0.039, AIC = 1396, $P = 0.57$). Percentage figures next to the variables indicate their explained variance ($R^2$). Figures on the arrows indicate standardized path coefficients and asterisks mark their significance; ***<0.001

wet and oxygen limited (Supplementary Fig. 3)[33,34]. We also note that in such soils, decreased turnover rate was associated with increased SOC concentration (Fig. 2b). Decreased microbial growth slows down the decomposition, leading to accumulation of plant and microbial organic matter. Evidence for higher microbial biomass accumulation in the acidic soils comes from the observation that the total microbial DNA pool in these soils was large relative to the amount of newly formed DNA. These results highlight the importance of considering both growth rate (the number of progeny produced per unit time) and efficiency (the number of progeny produced per unit of resource consumed) in examining microbial degradation and accumulation of organic matter[35].

Across all soils, specific respiration (qCO2) increased with decreasing SOC concentration (Fig. 2c), corroborating evidence from isotope tracer measurements of the interdependence of microbial growth efficiency and SOC accumulation. Microbial biomass measured as total soil DNA-C concentration[36,37] was higher in soils with higher SOC (Fig. 2d), highlighting positive relationship between microbial biomass and SOC[10,11,18]. We note that using DNA-C concentration as biomass proxy could lead to an underestimation of total microbial biomass and the absolute value of microbial CUE compared with approaches that employ other biomarkers. Despite a positive correlation between the potential activity of acetyl esterase[38] and SOC, a negative relationship was observed with enzyme production per unit biomass (Fig. 2e). Thus, at higher soil pH increasing CUE is associated with decreasing extracellular enzyme investment in substrate acquisition (Fig. 2a, e). This validates the physiological trade-offs theory suggesting that an increase in microbial enzyme investment in soils with depleted organic resources leads to reduced growth efficiency[20,25]. These relationships across soils varying in physico-chemical conditions and land use intensity provide a multitude of empirical evidence directly linking microbial ecophysiology to soil's C storage potential. We also observed significant relationships of taxonomic indicators like bacterial alpha diversity with SOC concentration (Fig. 2f); however, these were weaker than SOC correlations with key ecophysiological traits. This indicates that community-level physiological traits could be more strongly coupled to SOC accumulation than the assessed taxonomic diversity indices; this was further tested using structural equation modeling (SEM).

**Distinct SOC accumulation mechanisms across a pH threshold.** Empirical trends from recursive partitioning and generalized linear models suggest two distinct mechanisms of SOC accumulation across a soil pH threshold of ~6.2 for the sites assessed here: at sites above the threshold pH, greater microbial growth efficiency leads to increased microbial biomass that causes greater SOC accumulation;[11,18,25] and at sites below the threshold pH, acidity and wetness (Supplementary Fig. 3) leads to slower microbial growth rates (Fig. 2b) that limits decomposition of extant organic matter[39]. We sought to validate these mechanisms using path analysis of SEM[40]. Using a confirmatory approach, we aimed to test direct and indirect effects of microbial diversity and ecophysiology on SOC accumulation with the abovementioned distinct causal relationships. A latent biodiversity variable was derived as a common factor from independently assayed bacterial and fungal alpha diversity. Diversity here was simply used as a univariate predictor to represent these large multivariate molecular datasets. We analyzed a path structure based on a proposition that taxonomic diversity drives the trait response,[12,41] with higher metabolic efficiency causing greater microbial biomass and consequently leading to higher soil carbon accumulation. This model was valid only across higher pH soils (Fig. 3). At pH below the threshold value of 6.2, this model structure did not adequately reflect our dataset, disproving the potential chain of causality (Fig. 3) and indicating decoupling of relationships between CUE, microbial biomass, and carbon accumulation. The doctrine that growth efficiency and high biomass are associated with SOC accumulation is therefore not true in acidic soils. Here, microbial growth or turnover is a bigger constraint on decomposition resulting in higher organic matter accumulation, although communities exhibited low growth efficiency.

**Microbial physiological response to land use intensification.** The soils investigated here come from a range of land use types, from permanent grasslands with a long-term history of minimal agricultural improvement (low intensity) to intensively managed grasslands or arable croplands (high intensity). To test the effect of land use intensification on the ecophysiology of the soil microbial community, we used 21 locally paired low- and high-intensity land use contrasts from within this survey (Table 1, Supplementary Table 1). Given our focussed aim of targeting soils with long-term differences in land use intensity that typically

**Table 1 Characteristics of land use contrasts**

| Site pair ID | Site location | Low intensity contrast | | | High intensity contrast | | |
|---|---|---|---|---|---|---|---|
| | | Management | Soil pH | Soil C % | Management | Soil pH | Soil C % |
| *Type 1 land use effect* | | | | | | | |
| 1 | Hertfordshire | Unimproved grassland since 1949 | 6.4 | 3.7 | Intensive: arable | 6.4 | 1.6 |
| 2 | Hertfordshire | Unimproved grassland since 1949 | 6.6 | 2.8 | Intensive: arable | 6.9 | 1.5 |
| 3 | Hertfordshire | Unimproved grassland since 1900 | 6.8 | 4 | Intensive: arable | 7.5 | 1 |
| 4 | Bedfordshire | Unimproved grassland since 2002 | 7 | 1.5 | Intensive: arable | 7.2 | 1 |
| 5 | Oxfordshire | Unimproved grassland since 1990 | 6.9 | 2.9 | Intensive: arable | 7.4 | 2.3 |
| 6 | Oxfordshire | Unimproved grassland | 7.5 | 6.3 | Intensive: arable | 7.7 | 2.1 |
| 7 | Oxfordshire | Unimproved wet grassland | 7.6 | 15.7 | Intensive grassland | 7.6 | 8.8 |
| 8 | Cambridgeshire | Unimproved grassland | 7.6 | 7.4 | Intensive: arable | 7.9 | 4 |
| 9 | Devon | Unimproved grassland | 6.7 | 5.1 | Intensive: arable | 6.6 | 4.2 |
| 10 | Lancashire | Unimproved grassland since before 1980 | 6.7 | 6.3 | Intensive grassland | 6.9 | 5 |
| 11 | Wiltshire[a] | Unimproved calcareous grassland since before 1900 | 7.7 | 10.4 | Intensive: arable | 8 | 3.8 |
| *Type 2 land use effect* | | | | | | | |
| 12 | North Lanarkshire | Unimproved grassland since before 1985 | 5.8 | 6.9 | Intensive: arable | 6.4 | 3.8 |
| 13 | Devon | Unimproved wet grassland | 5.7 | 13 | Intensive grassland | 6.4 | 9.4 |
| 14 | Devon | Unimproved wet grassland | 5.3 | 13 | Intensive grassland | 6.4 | 5.7 |
| 15 | Buckinghamshire[b] | Unimproved grassland | 6.1 | 5.9 | Intensive: arable | 7.7 | 3.6 |
| 16 | Dorset | Unimproved grassland | 5.8 | 3.9 | Intensive grassland | 6.8 | 3.7 |
| 17 | Perthshire[a] | Unimproved grassland | 5.2 | 23.8 | Intensive grassland | 6.4 | 4.3 |
| *Type 3 land use effect* | | | | | | | |
| 18 | North Yorkshire | Unimproved grassland | 5.9 | 9 | Intensive grassland | 5.8 | 7.6 |
| 19 | Devon | Unimproved wet grassland | 5.3 | 9.8 | Intensive grassland | 5.7 | 10.4 |
| 20 | Devon[a] | Unimproved wet grassland | 5.8 | 17 | Intensive grassland | 5.8 | 4.3 |
| 21 | Dorset | Unimproved grassland | 5.6 | 5.2 | Intensive grassland | 6.2 | 3.7 |

Land use histories of the 21 paired low- and high-intensity contrasts and their mean soil pH and carbon concentrations
[a]Pairs used for metaproteomic analysis
[b]This contrast was not local; 4 km apart from each other

results in large changes in soil properties like SOC content, we did not collect detailed present-day plant properties like net primary productivity, plant diversity and chemistry of inputs. Instead, we assessed the impact of land use intensification on edaphic properties and used the observed changes in soil physico-chemical parameters to evaluate the microbial ecophysiological response. Land use intensification from more pristine grasslands to intensive agricultural systems generally tends to increase the soil pH and typically leads to decreased soil carbon concentration, reduced water retention and poorer soil structure[1,17]. We observed similar results (Fig. 4a–c). There were no coherent global effects of land use intensification on microbial ecophysiology, and the direction of change was not uniform across land use contrasts. Finer assessment pointed to characteristic trends that were linked to distinct mechanisms of microbial C cycling across the soil pH threshold described above. The effect of land use intensification depended on the quantity and direction of change in soil pH. We thus identified the following three categories of effects resulting from the various impacts of land use intensification on soil edaphic properties.

Type 1 effect of land use intensification: Plant productivity is highest in soils at the higher end of the pH spectrum, making such soils more favorable for agriculture; evidence suggests that the land cover area under intensive management tends to increase with an increase in mean soil pH[1]. In the high pH (>6.2) sites under investigation, intensification lead to a shift towards alkalinity, microbial community CUE generally decreased and growth rate increased (Fig. 4a), suggesting a shift towards copiotrophic life strategies with a wasteful metabolism. The resultant decrease in microbial biomass and increase in specific respiration ($qCO_2$) can be linked to SOC loss in intensive land use systems.

Type 2 effect of land use intensification: Liming acidic soils is a widespread agricultural practice used to improve the suitability of such soils for plant production[30,32]. In the low pH sites in our study where intensification lead to increased soil pH crossing the threshold (less intensive sites: pH < 6.2, adjacent more intensive sites: pH > 6.2), microbial CUE and turnover rates were both higher in more intensive soil systems, with a decrease in microbial biomass and increase in specific respiration (Fig. 4b). This suggests a shift in microbial physiology from the dominance of maintenance respiration in less-intensive acidic soils to increased growth and decomposition in higher-intensive pH soils. We postulate that SOC loss in such intensive soils is not a reflection of increased growth efficiency of microorganisms, but rather a result of the depletion in acid retardation of microbial growth indicated by increased growth rates that lead to increased organic matter decomposition[30].

Type 3 effect of land use intensification: This includes low pH sites (pH < 6.2), where the increase in soil pH on intensification does not cross the threshold of 6.2. Microbial CUE and turnover rates in both land use contrasts were systematically lower relative to the high pH sites, and intensification leads to a small decrease in both parameters (Fig. 4c). This is in line with our hypothesized SOC accumulation mechanism in low pH soils, where microbial growth and decomposition rates were lower (Fig. 2b) due to acidity and higher soil moisture content.

**Trade-offs in traits have consequences for SOC accumulation.** Soil metaproteomics was used to determine the physiological basis of microbial control on soil C transformations. This was achieved by analyzing functional changes across representative land use contrasts for each of the three types of land use intensification effects (Table 1). For the type 1 effect (both soils above

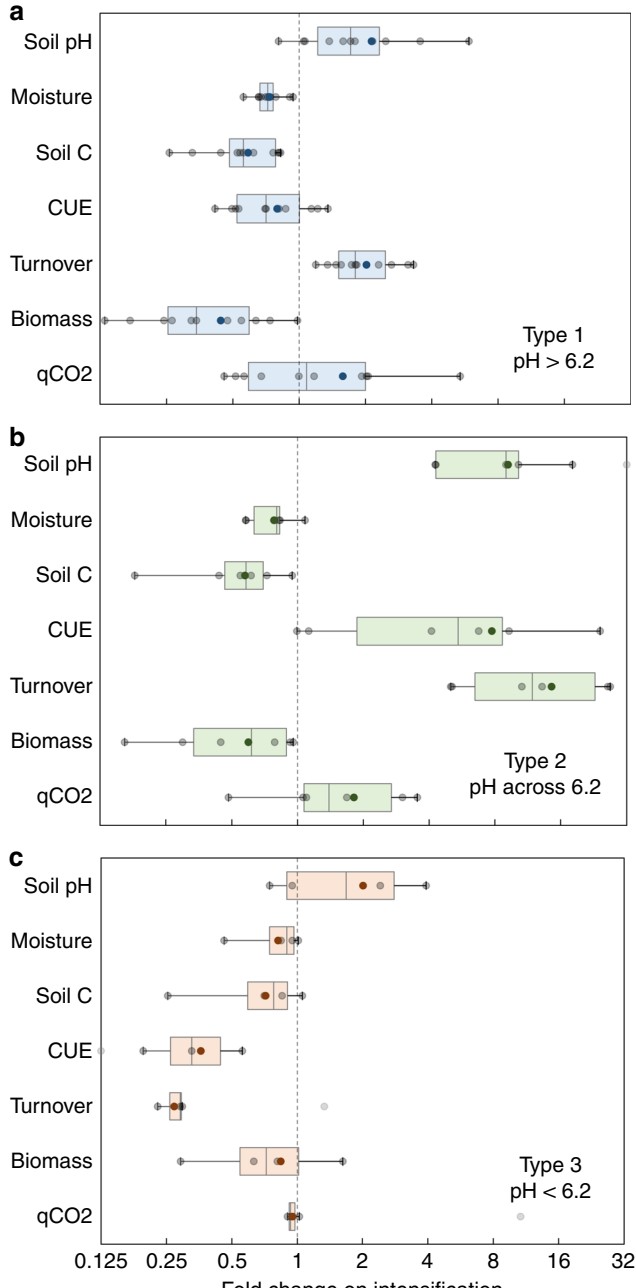

**Fig. 4** Land use impact on edaphic properties and microbial traits. Fold change of various edaphic properties and microbial physiological traits in the hypothesized categories of effects of land use intensification. Displayed here is the fold change in the measured parameter on land use intensification for 21 comparable local land use contrasts (no. of contrasts —type 1 (**a**): 11, type 2 (**b**): 6, and type 3 (**c**): 4; Table 1) that was calculated as the ratio of mean parameter values from low and high intensity treatments with three replicates each. Gray points represent means of fold change for each contrast, colored points represent the means across all contrasts, medians are marked by the vertical line inside the box, boxplots show quartile values for each parameter, and the whiskers extend to the highest and lowest values across all contrasts. Dotted line is the boundary indicating identical parameter values across the land use contrasts

pH 6.2), peptide profiling revealed that the more pristine land use systems have a significantly higher number of ABC transporters (KEGG level 1 class: environmental information processing) that included transport system substrate binding proteins for general

L-amino acid, branched chain amino acids, di/oligo peptides, ribose and glycerol, indicating a salvage pathway of substrate uptake and assimilation (Fig. 5a, b)[34,42,43]. On the contrary, lower availability of organic substrates in the intensive systems is reflected in lower relative abundance of ABC transporters and higher abundance of proteins linked to maintenance pathways, such as large and small subunit ribosomal protein synthesis, amino acid synthesis, purine metabolism (DNA-directed RNA polymerase subunits), and energy generating oxidative phosphorylation proteins like ATPase (Fig. 5b). Thus, the reduction in salvage pathways of resource acquisition, leads to a higher investment into non-growth maintenance activities like molecular turnover of proteins/enzymes for resource acquisition, and the energy generating pathways needed to fuel those activities. Microbial communities in high intensity soils also invest a higher proportion of resources and energy into stress alleviation indicated by a significant increase in stress protein indicators like Chaperonin GroEL (level 1 class: genetic information processing, level 3 class: RNA degradation; Fig. 5a, b)[17]. These molecular chaperones prevent protein aggregation by either refolding or degrading stress-induced misfolded proteins. The increased cellular investment into stress alleviation trades off with reduction in microbial growth efficiency measured as CUE (Fig. 4a). Thus, although soil communities in intensive soil systems at higher soil pH have greater turnover rate, low growth efficiency means a greater proportion of organic matter inputs are mineralized and lost in maintenance respiration (Fig. 4a). This ecophysiological understanding of microbial trade-offs further reiterates the significant microbial control on soil C cycling in high pH soils. In such intensive systems, soil management strategies that increase the microbial growth efficiency could help maximize organic matter accumulation and soil carbon storage. We suggest that cover crops and other restorative conservation approaches that are aimed at increasing plant carbon input into soils would over a period of time promote microbially mediated reiterative soil organic matter formation and decrease the resource limitation and water stress. Our functional assessments show that this would lower the metabolic constraints on microbial growth leading to a positive feedback on microbial growth efficiency, causing an increased channeling of substrates into biomass synthesis, thus fostering additional soil organic matter accumulation.

The microbial metabolic constraints in stressed environments such as arable croplands are also apparent in the type 2 effect of land use intensification (which increases the soil pH above the threshold value of 6.2) but the mechanisms here are distinct. Differences in the frequency and relative abundance of protein indicators under type 2 scenarios highlights several physiological adaptations. A decrease in acidity and moisture levels in the intensive land use system leads to increased microbial turnover (Fig. 4b) that is also reflected in increased investment in a number of metabolic pathways to fuel the need for energy and biosynthesis of the active microbial population (Fig. 5a). These included small and large subunit ribosomal protein synthesis, central metabolic pathways like glycolysis and TCA cycle, oxidative phosphorylation, purine metabolism, and amino acid biosynthesis (Fig. 5b). There was also an increase in ABC transporters, particularly the transport system for branched chain amino acids and phosphate in high intensity soils, as a result of higher substrate availability through increased degradation of extant organic matter[34,42,43]. Different stress proteins belonging to the families Chaperonin GroEL and molecular chaperone DnaK (level 3 class: RNA degradation) were identified as indicators of high and low intensity land use treatments but their relative abundances were significantly higher in the more acidic low intensity soils as an acid tolerance response[44]. The higher investment into stress alleviation in low intensity soils

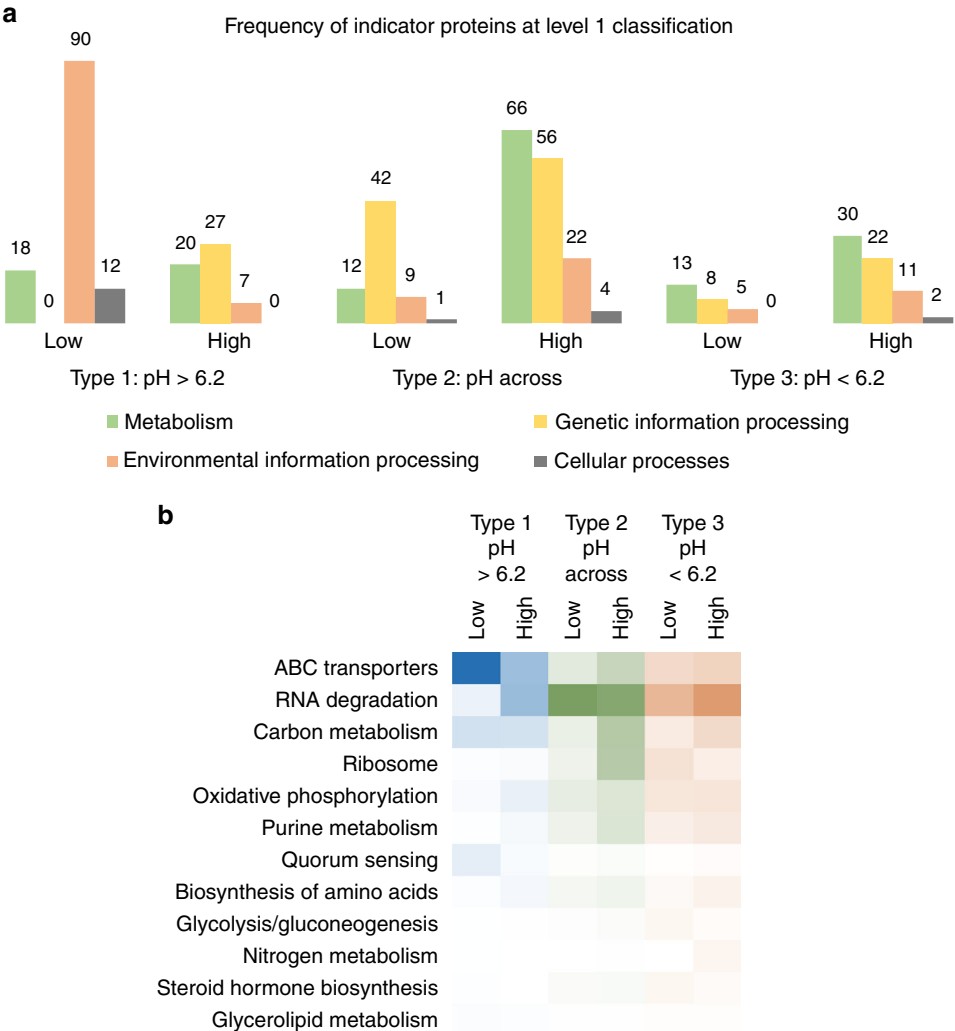

**Fig. 5** Functional indicators of land use change. Pairwise protein indicators of land use change in three representative land use contrasts each belonging to the hypothesized land use intensification effects: **a** frequency of protein indicators from each soil in KEGG main functional classification; **b** relative abundance of the most numerous protein indicators at level 3 classification, darker hues denote higher abundance. Each contrast consisted of a low and a high land use intensity treatment with three replicates each

along with the lower metabolic efficiency reiterates the trade-offs in microbial functional traits.

In the type 3 intensification effect, where both contrasts were below the threshold pH value of 6.2, pairwise protein indicator analysis presented fewer functional indicators of land use intensification since the two systems were similar (Fig. 5a). Key indicators were stress proteins that were more abundant in high-intensity soils; it is worthwhile to note that these were global indicators of intensive land use systems across all three land use contrasts, and have been previously reported in agricultural systems[17]. As highlighted earlier, the microbial controls on soil C cycling processes were much lower in low pH soils, as evidenced by lower microbial turnover rates. However, soil microorganisms continued to respire possibly using non-plant C sources like direct C fixation[34], although we did not find any evidence from proteomic analyses about alternative C respiration redox strategies. Observations emphasizing that microbial physiological trade-offs are key determinants of the microbial contribution to soil organic matter formation thus need to be further tested to fully understand the biochemical basis of soil carbon accumulation. Nevertheless, we present strong evidence of such trade-offs in microbial traits being important in near-neutral soils, which is where the majority of productive agriculture occurs.

## Discussion

We established landscape-scale empirical links between key microbial ecophysiological traits and soil C concentration supporting the central role of microorganisms in belowground carbon cycling. Results show that an efficient microbial physiology with a greater proportion of substrate allocated to biosynthesis manifests in the increased ability of such communities to store C in near-neutral pH soils. Trade-offs in microbial physiological traits determine the proportion of microbial organic C investment into biosynthesis. Growth and biosynthesis decline in scenarios of stress and resource limitation when cellular investment is far higher in traits focussed on stress tolerance and resource acquisition. We discern two distinct mechanisms of soil carbon accumulation across a pH threshold of 6.2 for these soils: at higher pH (>6.2), an efficient substrate metabolism leads to increased SOC accumulation; and in acidic wet environments (pH < 6.2), abiotic factors limit microbial growth and decomposition causing accumulation of SOC. This evidence supports the use of soil pH as an integrated proxy of land use change, parent material and climate[39] to determine the site-specific effects of land management strategies on SOC accumulation. The mechanisms highlight the significance of microbial ecophysiological controls on soil organic matter accumulation in high pH soils. Here, less-intensive land

management practices have greater potential for soil carbon storage through increased microbial growth efficiency that causes greater channeling of substrates into biomass synthesis. Intensification in low pH soils leads to alleviation of acid-related retardation of microbial growth and organic matter degradation, leading to large losses of carbon through microbial decomposition. In these systems, preserving equivalent amounts of organic C would involve managing the abiotic C-accumulating factors, like acidity and wetness, whilst enhancing plant production. We thus highlight the importance of including physiological attributes of soil microorganisms in designing restorative land management strategies aimed at mitigating losses of soil C by intensive agricultural practices.

## Methods

**Sampling regime.** Soil samples were collected from 56 geographically distributed sites that differed in their edaphic properties (Fig. 1). From these sites, 21 local contrasts of land use intensity with relevant historical knowledge of land management scenarios were chosen to study the effect of land use intensity on microbial functionality (Table 1). Research experiments, working farms or farms under management by local wildlife trusts were targeted to provide two nearby fields of differing long-term management intensity. The low-intensity sites were generally permanent unimproved grasslands, often with low-density grazing. The high-intensity sites were mostly arable fertilized croplands or intensive grasslands. The subset of 21 local contrasts were selected based on the criteria that the higher-intensity treatment had a significant reduction in organic matter and an increase in pH reflecting nationwide trends of intensification effects (Table 1, Supplementary Table 1). From each of the identified sites at a single time point, three spatially dispersed soil cores (5 cm diameter, 15 cm deep) were sampled to capture the natural spatial variability at each site. After all the visible roots were removed, aliquots of the homogenized soil were frozen at −20 °C until the following functional analyses.

**Soil chemistry.** Loss on ignition (LOI), C, N, pH, moisture, and clay content were assessed in replicated soil samples from each site using standard protocols. Soil total C concentration strongly correlated with LOI ($R^2 = 0.93$), with the only deviations being for the calcareous soils with high inorganic C. For soil from these sites, total C was predicted based on the LOI–C regression model for the non-calcareous sites. Soil pH was measured in deionised water and moisture content was determined by the gravimetric method.

**Microbial respiration.** For microbial basal respiration measurements, an aliquot (1 g) of the field-moist soil was placed in a 10-mL glass vial and incubated overnight (for ~16 h) in the dark at room temperature (21 °C) without manipulating the moisture levels. Respired $CO_2$ collected in the headspace was measured using a gas chromatography isotope ratio mass spectrometer (GC-IRMS, Delta + XL, Thermo Fisher Scientific, Germany) coupled to a PAL autosampler (CTC Analytics) with general purpose (GP) interface (Thermo Fisher Scientific, Germany). Following the basal respiration measurement, vials with soils were opened and incubated in the dark for 8 h before adding 100 μL of [13]C-labeled DOC solution (0.13 mgC). The filter-sterilized solution was prepared from [13]C-labeled powdered plant leaf litter, containing a range of compounds varying in their decomposability. Leaf litter was produced by growing a temperate herb in a [13]$CO_2$ atmosphere (~1 atom% [13]C at 400 ppm)[45]. Respiration measurements were repeated following the same incubation procedure as mentioned above to obtain the proportion of DO[13]C in respired $CO_2$.

**Microbial biomass, CUE and turnover rate.** Soil microbial total DNA-C concentration was used as a proxy for biomass C; DNA extraction was carried out on a soil aliquot of 0.25 g using PowerSoil-htp 96-well soil DNA isolation kit following the manufacturer's instructions (MO BIO Laboratories, UK). Another set of identical DNA extraction was performed following addition of 25 μL of DO[13]C solution and overnight (16 h) incubation in the dark at field moisture capacity. Both extracts were analyzed in the size-exclusion chromatographic (SEC) mode on a liquid chromatography isotope ratio mass spectrometer LC-IRMS (HPLC system coupled to a Delta + XP IRMS through an LC IsoLink interface; Thermo Fisher Scientific, Germany)[46]. This allowed us to obtain the DNA-C content and the proportion of DO[13]C in microbial DNA from soils with and without substrate addition. Microbial CUE was estimated as DNA-[13]C/(DNA-[13]C + $\Sigma CO_2$-[13]C), where $\Sigma CO_2$-[13]C is the cumulative DO[13]C lost during respiration. Microbial turnover rate (synonymously referred to as growth rate) was calculated as DNA-[13]C/DNA-C.

**Taxonomic diversity.** An aliquot of the DNA was used to perform 16 s rRNA gene- and ITS region-tagged amplicon sequencing using Illumina MiSeq platform to estimate the bacterial and fungal taxonomic diversity, respectively, using

previously reported dual index methods[47,48]. Raw sequences from amplicon sequencing were quality filtered, merged and clustered to generate OTU's at 97% sequence similarity. Richness was calculated on rarified data (2000 reads) using the R vegan library.

**Extracellular enzyme assay.** The potential activity of the enzyme acetyl esterase was estimated with the common assay protocol using fluorigenic substrates. The enzyme acetylxylan esterase is the most relevant acetyl esterase in soil; it belongs to the family of hydrolases that catalyzes the deacetylation of xylans (the main hemicelluloses in hardwoods and annual plants) and was chosen as it is involved in the initial stages of depolymerization of complex plant substrates[38]. Briefly, the soil washes were obtained using 1.5 g of homogenized soil shaken in 20 ml of deionized water[49]. The resultant slurry was used to perform esterase activity assay using 4-Methylumbelliferyl acetate esterase substrate. The reaction was performed for 3 h at 28 °C, with one fluorometric measure every 30 min (BioSpa 8 Automated Incubator). Fluorescence intensity was measured using a Cytation 5 spectrophotometer linked to the automated incubator.

**Metaproteomics.** For proteomic analysis of microbial communities, three replicated samples were used from both high- and low-intensity land use, consisting of six sites, each contrast representing one of the three types of hypothesized land use intensification effects (Table 1). A total of 5 g of soil was used for protein extraction (with two technical replicates) using the SDS buffer–phenol extraction method[45]. The protein extract was purified using 1D SDS-PAGE and the resultant product was subjected to tryptic digestion. Proteolytically cleaved peptides were separated prior to mass spectrometric analyses by reverse-phase nano HPLC on a nano-HPLC system (Ultimate 3000 RSLC nano system, Thermo Fisher Scientific, San Jose, CA, USA) coupled online for analysis with a Q Exactive HF mass spectrometer (Thermo Fisher Scientific, San Jose, CA, USA) equipped with a nano electrospray ion source (Advion Triversa Nanomate, Ithaca, NY, USA). Raw data from the MS instrument were searched using Proteome Discoverer v1.4.1.14 (Thermo Fisher Scientific) against a FASTA-formatted database (protein coding sequences of bacteria, fungi and archaea, Uniprot 05/2016) using the SEQUEST HT algorithm. Database searches were performed with carbamidomethylation on cysteine as a fixed modification and oxidation on methionine as a variable modification. Enzyme specificity was selected to trypsin with up to two missed cleavages allowed using 10 ppm peptide ion and 0.02 Da MS/MS tolerances. Only peptides with a false discovery rate (FDR) < 1% estimated by Percolator[50] and only rank 1 peptides were accepted as identified. Unipept v3.2[51] was applied to assign proteins to their phylogenetic origin. GhostKoala[52] and KEGG classifier were used for functional annotation.

**Regression analyses.** Statistical analyses were performed under the R environment software 2.14.0[53]. Regression tree analysis for recursive partitioning of microbial CUE data by edaphic properties that included soil pH, moisture, total C and N concentration, C:N ratio and clay content was performed on the entire dataset (56 sites) using the rpart package. Here, all three replicates from each site were treated independently to account for spatial non-independence. Slope failure test or piecewise regression for the CUE–soil C linear regression was performed by recurrent movement of the soil pH window over a range of 1.4 units, with increments of 0.1 units. The threshold pH value was determined as the last pH unit before the $R^2$ of the CUE–soil C regression dropped dramatically (Supplementary Fig. 2). Following threshold pH determination, the data were segregated into low (<6.2, $n = 50$) and high (>6.2, $n = 113$), and generalized linear models of various physiological traits and soil C were run for both datasets separately.

**Structural equation modeling.** This was applied to test direct and indirect effects of microbial diversity and ecophysiology on soil carbon accumulation by organizing the dataset into a path relation network. Using a confirmatory approach, we aimed to test the maximum likelihood of data fit to the hypothesized path model inferring microbial taxonomy and function influences soil C accumulation. Microbial functional traits that were included as predictor variables were CUE and biomass, and a latent variable for microbial taxonomy was generated using bacterial and fungal alpha diversity. The fit of the path model and structural relationships with data were verified using SEM analysis conducted with the lavan R package[54]. The most parsimonious model was identified by non-significant X2 tests ($P \geq 0.05$), low Akaike Information Criterion (AIC), low Root Mean Square Error of Approximation index (RMSEA ≤ 0.1), low Standardized Root Mean Square Residual index (SRMR ≤ 0.1) and high Comparative Fit Index (CFI ≥ 0.90).

**Analysis of treatment effect size.** To assess the effect of land use intensification, we determined the quantity and direction of change in the measured microbial traits across the 21 chosen land use contrasts (42 sites with three replicates at each site). Fold change on intensification was calculated for each contrast as the ratio of mean trait values from low- and high-intensity treatments. Following the failure to elucidate any global patterns, we segregated the data based on our mechanistic understanding of soil C cycling into categories of contrasts that fall above and below the hypothesized threshold soil pH and into a third category of contrasts

where land use intensification leads to soil pH shift from below to above the threshold value (Table 1, Fig. 4a–c).

**Protein indicator analysis.** Pairwise Indicator Species Analysis was used to identify the protein functions (from soil metaproteomics) that were significantly associated with low- and high-intensity land use treatment for each contrast[55]. This was implemented within the R library labdsv (http://ecology.msu.montana.edu/labdsv/R). The IndVal score for each protein is the product of the relative frequency and relative average abundance within each land use treatment, and significance was calculated through random reassignment of groups (1000 permutations).

## Data availability

The authors declare that the data supporting the findings of this study are available within the article and its Supplementary Information file, and from the corresponding author on request. The mass spectrometry proteomics data generated during the current study are available in the ProteomeXchange Consortium via the PRIDE partner repository with the dataset identifier PXD010526.

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

## Acknowledgements

A.A.M. has received funding from the European Union's Horizon 2020 research and innovation program under the Marie Skłodowska–Curie grant no. 655240 and all other UK-based authors were funded by the UK Natural Environment Research Council under a Soil Security Programme grant (NE/M017125/1). We also acknowledge Kathleen Eismann for her help in sample preparation for metaproteomic analysis; Heiko Moossen, Petra Linke and Steffen Ruehlow for assistance with stable carbon isotope analyses and all the land owners who provided sampling access.

## Author contributions

A.A.M., J.P., K.M.B., T.G., and R.I.G. designed the research; A.A.M., T.G., and S.C. performed the stable isotope analyses; N.J. performed metaproteomic analysis; T.G. and H.S.G. performed the taxonomic diversity assessments; J.P. and T.G. performed the enzyme assays; A.A.M. and J.P. performed statistical analyses; A.A.M., J.P., K.M.B., T.G., J.M.P., K.E.M., M. vA., A.B., I.M.C., J.W., R.F.P., N.O., and R.I.G. were involved in site selection, soil sampling, and processing; N.J., G.G., and R.I.G. contributed new reagents and analytical tools; A.A.M. drafted the manuscript and all authors were involved in critical revision and approval of the final version.

## Additional information

**Competing interests:** The authors declare no competing interests.

