## [Peer Review File · Nature Communications]

Reviewers' comments:

Reviewer #1 (Remarks to the Author):

The authors analysed microbial traits across a wide range of different soils and present a concept of how soil pH affects mechanisms of soil carbon storage. We are still a long way from a satisfactory understanding of microbial carbon cycling, its controls and how it exactly relates to soil carbon storage. Given the current pressure understanding and predicting soil carbon storage, the topic is not only timely but also of interest to a broader audience, ranging from microbiologist, soil and agricultural scientists to earth system scientists.

The manuscript is well structured and written. Statistical analyses of the data presented are appropriate and support the author's conclusions. The combination of the analysis of microbial carbon use efficiency together with proteomics give interesting insights into physiological traits of microbial communities. This study significantly contributes towards a better understanding of the controls of environmental factors, here soil pH, on underlying mechanisms related to soil carbon storage and provides a framework for the integration of microbial physiological traits into the mechanisms of soil carbon storage in the context of land management strategies.

In the present study, I see one potential problem of the applied method that could result in biased estimates of microbial carbon use efficiency: no conversion factor to extrapolate from DNA to microbial biomass C was applied. The authors state that they used DNA as a proxy for biomass. However, the ratio of biomass C to DNA can strongly vary between soils, which, e.g., could be due to differences in microbial community structure. Fungi have a higher biomass C to DNA ratio, which generally is also more variable than that of bacteria (Leckie et al., 2004). It has been shown that this conversion factor can range between 2 and 60 for different soils (e.g., Joergensen & Emmerling, 2006; Anderson & Martens, 2013; Fornasier et al., 2014). Differences in this conversion factor could be especially relevant in the present study, because a wide range of different soils was used, that are more likely to differ in this factor. So, I was wondering if the observed relationship of microbial carbon use efficiency or the lack thereof hold true if a soil-specific conversion factor is applied to estimate microbial biomass C production from newly formed DNA.

Other comments:

Regression tree analysis (Figure S1): Given that soil N was determined for all samples, I was wondering why soil C:N was not included in this analysis? Based on what we know about microbial carbon use efficiency and its control, soil C:N may contribute explaining some of the variation in the data set.

Figure S3: Give unit or standardization for soil moisture.

References:

- Anderson, T.-H., Martens, R. DNA determinations during growth of soil microbial biomasses. *Soil Biol. Biochem.* 57, 487-495 (2013).
- Fornasier, F., Ascher, J., Ceccherini, M.T., Tomat, E., Pietramellara, G. A simplified rapid, low-cost and versatile DNA-based assessment of soil microbial biomass. *Ecol. Indic.* 45, 75-82 (2014).
- Joergensen, R.G., Emmerling, C. Methods for evaluating human impact on soil microorganisms based on their activity, biomass, and diversity in agricultural soils. *J. Plant Nutr. Soil Sci.* 169, 295-309 (2006).
- Leckie, S.E., Prescott, C.E., Grayston, S.J., Neufeld, J.D., Mohn, W.W. Comparison of chloroform fumigation-extraction, phospholipid fatty acid, and DNA methods to determine microbial biomass in forest humus. *Soil Biol. Biochem.* 36, 529-532 (2004).

Reviewer #2 (Remarks to the Author):

The theme of this manuscript is very interesting and timely. However, it is hard to assess if the authors could, in fact, test their hypothesis given that the table S1 reporting field sites characteristic is in a zip file I was unable to open, and the method section misses important details (see specific comments below). Methods need to be written in more details, and site characteristics need to be presented in a smaller easy to access table, before this manuscript can be fully reviewed.

My main concern of this study is that the authors set to assess the effects of "land use" on microbial traits but, to my understanding, they only assess the effects of pH, moisture, total C and clay. These are very few edaphic properties, which are also often correlated among each other. In addition clay is not expected to change with land use. I would have anticipated plant input amounts, chemistry and distribution between above and below ground, as well as other management inputs or disturbance to be other key drivers, but they are not mentioned here. And again it's hard to assess the role these other factors may have had since the data on sites characteristics is so hard to access. In my opinion, the authors should either broaden their analyses to other key factors changing with land use, and likely to affect microbial traits, or drop the centrality of land use in their study and center it on assessing the effect of soil pH on microbial CUE.

Also, it's questionable that a "true" CUE can be assessed by applying the same easily decomposable substrate to all soils. It is likely that soils having microbial community adapted to use that kind of input would show a higher CUE than soils with communities developed to use very different or less labile substrates. Not considering or using the native inputs in the assessment of microbial CUE is a major drawback of this study.

Line specific comments:

Line 40: I would argue that in recent years we have actually made a significant progress in that direction and many important research articles and opinion papers were published on the formation and persistence of SOM. As the authors know and cite below. I suggest deleting this sentence
Figure 2: Specify that CUE and turnover rate refer to microbial

Line 314: It is not clear what the authors did here. If they collected many soil cores per sites and composited them into three replicate samples per site, that is OK, but they need to specify how the three areas where cores were collected and composited separate from each other. If only 3 soil cores were collected and composited in one sample per site would be an insufficient sampling strategy, missing to capture the natural spatial variability at each site.

Line 319: Again this is not clear, and stems from my comment above. Are those three through field replicates of soil samples obtained from the combination of several soil cores, or are they laboratory triplicates of one homogenized soil sample from three soil cores? The authors need to clarify. The first option is OK, the second is not acceptable.

Line 321: Please clarify this correction factor. Of course they did not correlate In calcareous soils because of the presence of significant amount of soil inorganic carbon (SIC). Was SIC quantified, and subtracted from the total C, or were all analyses conducted on total C?

Line 321-323: Were soil maintained at a standard moisture? Differences in field moisture would overwhelm any other difference as control of soil respiration. It is very important that the authors clarify soil moisture levels during incubation.

Line 330: How long after the addition of the labeled DOC? In which conditions were the soil stored for this incubation?

Line 332: Were fresh field soil used for this DNA extraction?

Line 334: Why was this other incubation performed, couldn't the soils from the previously described incubation used for DNA extractions?

Line 340: How could this equation be applied if (assuming I understood well) the respiration and DNA extractions were performed on different sets of soils? Methods need to be much better described to be assessed for correctness.

Line 378: soil pH, moisture, total C and clay are very few edaphic properties, which are also often correlated between each other. What I find surprising in this study is the total lack of data relative to plant input and chemistry, which are surely largely affected by the above factors, in particular pH, as well as management and that can well be the primary factors affecting microbial traits. What does total soil C refers to, % total C (ignoring carbonates?) again see my comment above, what was LOI used for?

Line 389: To my understanding, only %C and not soil C stocks were determined and used for the analyses. So the relation is with "soil C density" not storage

Reviewer #3 (Remarks to the Author):

Malik et al. produce a set of results that are intriguing and compelling. They are intriguing because they suggest very different roles for microbes in more acidic vs. more neutral soils in soil carbon cycling. Specifically, that in more neutral soils that aggregate properties of the microbial communities (e.g. CUE, turnover) drive soil C formation rates. Whereas in more acidic soils these same microbial variables appear detached from soil C turnover, with acidity instead limiting decomposition rates, meaning that soil C stocks are primarily controlled by abiotic factors. The results are compelling because they draw together a range of different methods for assessing the soils, from state-of-the-art (e.g. proteomics) to well-established and conventional methods (e.g. soil respiration potentials), and the data from each appear to line up to support the authors' interpretation of the patterns observed. Given the clever way the findings from the different methods were woven together to form logical and interesting arguments in the text, combined with the intriguing finding that microbes might act as direct controls on soil C dynamics at more neutral pH and yet be disconnected at more acid pH values, I read the paper favorably. However, I also had numerous questions that need to be addressed before the work can be fully evaluated. My assessment was likely hindered by the fact the Data S1.xlsx file would only download as a corrupt zip file, because many of my questions related to replication and sampling design. However, such information should not only be in supplementary material but readily available by reading, for example, figure legends. As well as rectifying these omissions, I'd ask the authors to consider the following points:

1. pH is on a log scale. Do microbes see pH or do they see hydrogen ion concentration? If you repeated your analysis with pH back-transformed to H⁺ ion concentration, would any of your conclusions change. For example, would the threshold value remain as pH 6.2?
2. Related to pH, I could not find in the Methods how you calculated pH. Given its prominence in the findings, it seems you really should detail this and clarify, for example, if taken in water or in CaCl₂, which would then alter the meaning of 6.2 for people.
3. Figures 2 and 4 are key data figures for establishing a robust empirical basis for the arguments you weave together. But I could not discern the n: please state this on the figure. You do state elsewhere that you had 56 sites and took 3 replicate samples per site, but in Fig. 2 for example you clearly have many more than 56 blue-circle symbols, leaving me confused. So are you treating each replicate from a site as independent?
4. Similarly, I do not know the n for your statistical models. These should also be stated. And please clarify how are treating the 3 replicate samples per site – it seems you might average them or better still have a random effect to account for potential spatial non-independence.
5. Your statement on line 102 that you treated SOC concentration as a proxy for soil C storage has huge consequences for the arguments that you build. I would argue that you need to carefully go through your paper and much better caveat your conclusions/ language given this assumption. It appears from your methods that you have a single soil collection, and hence a single snapshot of microbial variables from which you make many inferences about the consequences for much longer term soil C dynamics. Although your arguments are logical and interesting, they should be much more cautious. For example, you have an observational study. How do we then discern whether it is the

microbes affecting soil C, or the soil C driving the microbial community properties? Further, your arguments rely on greater SOC formation at >6.2 pH but with greater microbial efficiency and turnover would we also not expect greater microbial-mediated decomposition. So what does that mean for soil C stocks (which you do not actually measure): will more gain be balanced by more loss? And what exactly is the mechanism of acid protection of soil C, and if it's so important why does microbial biomass correlate strongly with soil C concentration without any change for the pH threshold? Overall, I like the paper and the approach and findings are interesting and compelling; but the over-extrapolation of the results was a strong detraction for me. You do this throughout the script, and an interpretation better constrained by the data would help the paper a great deal, and not lessen the value/ impact of the work.

6. Line 139 – strange that you would even think that taxonomic diversity indices would be more strongly linked than community physiological traits. This part seems a little bit of a straw man. Can you back that expectation up with citations, and maybe also present the counter-arguments that physiological traits should matter more?

7. Line 155 – can you give a citation for why you think taxonomic diversity would drive the trait response?

8. Line 160 – this is clearly a key argument but I do not follow the logic. Why does a weak link between CUE and biomass, but a strong link from biomass to soil C concentration, suggest abiotic factors dominate?

9. You mention a few a times that pH is an integrated proxy of conditions. Yet many of your arguments are premised on acidity being a stress. Which is it? Is pH acting directly, or via mechanisms such as control on P availability? Please ensure the inferences are internally consistent.

10. The supplementary figures need more explanation to be helpful. For example, in the regression tree pH 6.2 is not identified as a threshold. And what criteria are used to establish slope failure in the coefficient/ r^2 figure? It would seem that the r^2 and coefficient was still pretty strong for where you placed the threshold.

11. The land use intensification subset is intriguing but given the absence of data on n, it was not clear how this subset really related to the 56 full sites, how many observations you used when you got down to 21 paired locations, and so on. Without such information, your whole design remains somewhat of a mystery.

Response to reviewer comments

Reviewer #1

The authors analysed microbial traits across a wide range of different soils and present a concept of how soil pH affects mechanisms of soil carbon storage. We are still a long way from a satisfactory understanding of microbial carbon cycling, its controls and how it exactly relates to soil carbon storage. Given the current pressure understanding and predicting soil carbon storage, the topic is not only timely but also of interest to a broader audience, ranging from microbiologist, soil and agricultural scientists to earth system scientists.

This is right. We need more empirical as well as mechanistic approaches to tease apart the biotic and abiotic controls on soil C storage.

The manuscript is well structured and written. Statistical analyses of the data presented are appropriate and support the author's conclusions. The combination of the analysis of microbial carbon use efficiency together with proteomics give interesting insights into physiological traits of microbial communities. This study significantly contributes towards a better understanding of the controls of environmental factors, here soil pH, on underlying mechanisms related to soil carbon storage and provides a framework for the integration of microbial physiological traits into the mechanisms of soil carbon storage in the context of land management strategies.

We thank the reviewer for their affirmation.

In the present study, I see one potential problem of the applied method that could result in biased estimates of microbial carbon use efficiency: no conversion factor to extrapolate from DNA to microbial biomass C was applied. The authors state that they used DNA as a proxy for biomass. However, the ratio of biomass C to DNA can strongly vary between soils, which, e.g., could be due to differences in microbial community structure. Fungi have a higher biomass C to DNA ratio, which generally is also more variable than that of bacteria (Leckie et al., 2004). It has been shown that this conversion factor can range between 2 and 60 for different soils (e.g., Joergensen & Emmerling, 2006; Anderson & Martens, 2013; Fornasier et al., 2014). Differences in this conversion factor could be especially relevant in the present study, because a wide range of different soils was used, that are more likely to differ in this factor. So, I was wondering if the observed relationship of microbial carbon use efficiency or the lack thereof hold true if a soil-specific conversion factor is applied to estimate microbial biomass C production from newly formed DNA.

This is a valid concern – there is no simple conversion factor to estimate total microbial biomass C from DNA C concentrations. In the soil literature, biomass has often been measured using conversion factors based on measurements of chloroform fumigation extracts (CFE), assuming the method as a standard to measure total microbial biomass. However, some of our

published papers comparing flow of C into different cellular fractions (DNA, RNA, PLFA, CFE and its molecular size classes) highlight that CFE is incomplete, highly variable, selective and only represents certain fractions of microbial cells (Malik et al., 2016, 2015). Another recent study found that CFE microbial biomass does not correlate with other microbial measures like ATP, flow cytometry cell numbers, qPCR and PLFA (Zhang et al., 2017). Other culture-based conversion factors have also been shown to be highly variable and do not reflect the soil microbial biomass measures. Thus, we are yet to have a standard method that truly measures total soil microbial biomass. For this reason, we refrained from using a conversion factor.

However, to further support our data, we also found strong correlations of DNA C concentrations with bacterial cell abundance estimated using flow cytometry and with fluorescein diacetate (FDA) hydrolysis which indicates the potential overall microbial activity. This made us believe that DNA C concentrations could be used as a biomass proxy, without necessitating conversion to absolute biomass. In addition, measuring ¹³C incorporation in DNA is a more appropriate way of measuring growth efficiency (CUE) than CFE or PLFA as new DNA production explicitly infers growth/reproduction in microorganisms, whereas variability in other cellular constituents may arise from maintenance processes.

With respect to the reviewer's argument about fungi having higher and variable biomass C to DNA ratio, we agree with the comments, though do not think this is of consideration here. We note that fungi only contribute minimally to the soil DNA pool (1.8-3% based on whole genome metagenomic evidence- Malik et al., 2017); and soil proteomic analysis from the present study also found that fungal proteins show low abundance (0.4-1.8%) in the total protein pool.

Other comments:

Regression tree analysis (Figure S1): Given that soil N was determined for all samples, I was wondering why soil C:N was not included in this analysis? Based on what we know about microbial carbon use efficiency and its control, soil C:N may contribute explaining some of the variation in the data set.

We have now included nitrogen concentration and C:N ratios in the regression tree analysis but as expected both don't feature in the resultant tree. Method details have been modified accordingly.

Figure S3: Give unit or standardization for soil moisture.

Amended

References:

Anderson, T.-H., Martens, R. DNA determinations during growth of soil microbial biomasses. *Soil Biol. Biochem.* 57, 487-495 (2013).

Fornasier, F., Ascher, J., Ceccherini, M.T., Tomat, E., Pietramellara, G. A simplified rapid, low-cost and versatile DNA-based assessment of soil microbial biomass. *Ecol. Indic.* 45, 75-82 (2014).

Joergensen, R.G., Emmerling, C. Methods for evaluating human impact on soil microorganisms based on their activity, biomass, and diversity in agricultural soils. *J. Plant Nutr. Soil Sci.* 169, 295-309 (2006).

Leckie, S.E., Prescott, C.E., Grayston, S.J., Neufeld, J.D., Mohn, W.W. Comparison of chloroform fumigation-extraction, phospholipid fatty acid, and DNA methods to determine microbial biomass in forest humus. *Soil Biol. Biochem.* 36, 529-532 (2004).

We thank the reviewer for sharing these references.

Reviewer #2

The theme of this manuscript is very interesting and timely. However, it is hard to assess if the authors could, in fact, test their hypothesis given that the table S1 reporting field sites characteristic is in a zip file I was unable to open, and the method section misses important details (see specific comments below). Methods need to be written in more details, and site characteristics need to be presented in a smaller easy to access table, before this manuscript can be fully reviewed.

We've now added more details in the methods and results section. We've also added tables of site descriptions in the main manuscript and Supporting Info. Supplementary file with site description was a small 38KB Excel file, we assume something went wrong with the file download system.

My main concern of this study is that the authors set to assess the effects of "land use" on microbial traits but, to my understanding, they only assess the effects of pH, moisture, total C and clay. These are very few edaphic properties, which are also often correlated among each other. In addition, clay is not expected to change with land use. I would have anticipated plant input amounts, chemistry and distribution between above and below ground, as well as other management inputs or disturbance to be other key drivers, but they are not mentioned here. And again, it's hard to assess the role these other factors may have had since the data on sites characteristics is so hard to access. In my opinion, the authors should either broaden their analyses to other key factors changing with land use, and likely to affect microbial traits, or drop the centrality of land use in their study and centre it on assessing the effect of soil pH on microbial CUE.

In this manuscript, we first set out to present the empirical relationships between microbial physiological traits and soil properties. We did not collect detailed plant-related data like NPP, plant diversity and chemistry of inputs, given our focussed aim of targeting soils with long term differences in land use intensity. Such long-term practices typically result in large changes to soil organic matter contents, which are likely to be independent of the present-day

plant properties (for instance the type of crop grown in an intensive arable field around the time of sampling is unlikely to influence the soil properties and communities as much as the longer-term effects of arable cropping practices). Therefore, for assessing the specific impacts of intensification on soil microbial processes we chose not to direct resources into a detailed plant community characterisation. Instead, we aimed at assessing the effect of land use intensification on edaphic properties and its consequences on microbial physiology. Once we found linkages between soil properties and microbial ecophysiological traits, we set out to analyse if the direction of change is uniform across all the land use contrasts. We observed that it wasn't, and the effect of land use intensification depended on the quantity and direction of change in soil pH. We have now included details about our approach in the manuscript (line 183-200, line 434-442). Two new tables, one in the main MS and the other as Supplementary Info provide more information about sites and their land management histories to better interpret the results.

Also, it's questionable that a "true" CUE can be assessed by applying the same easily decomposable substrate to all soils. It is likely that soils having microbial community adapted to use that kind of input would show a higher CUE than soils with communities developed to use very different or less labile substrates. Not considering or using the native inputs in the assessment of microbial CUE is a major drawback of this study.

We concur that we are not measuring the "true" CUE, but a standardised assay of CUE which is all that is currently practically achievable. However, we note that our method is a marked improvement in comparison to conventional methods. Previously, CUE has been measured without the use of stable isotope tracing approaches or in cases where tracers have been used labile substrates like glucose were generally added. Our use of plant leaf litter DOC is likely a closer reflection of in situ substrates as the DOC should contain a range of compounds varying in their complexity and decomposability allowing us to mimic in situ conditions (added in line 362). We also add very small amounts of DOC (0.13mg/g soil; roughly 0.1-0.6% of native OC) to minimise disturbing the system. In addition, our technique measures incorporation into microbial DNA using a size exclusion chromatography-based LC-IRMS method that is state-of-the-art that facilitated improved measurement efficiency and accuracy. Adding isotopically labelled native inputs from 56 sites was not feasible, hence we preferred to use a "common garden" approach. All this makes us believe that in spite of the methodological shortcomings that exists, we were able to measure and compare CUE values across soils and treatments.

Line specific comments:

Line 40: I would argue that in recent years we have actually made a significant progress in that direction and many important research articles and opinion papers

were published on the formation and persistence of SOM. As the authors know and cite below. I suggest deleting this sentence

Yes, we agree. We have made some progress but as reviewer 1 also mentions in their response, we are still a long way from a satisfactory understanding of the microbial controls on soil carbon cycling and how it relates to soil carbon storage. We've changed the phrase from "we do not currently understand" to "we do not sufficiently understand".

Figure 2: Specify that CUE and turnover rate refer to microbial

We've now specified the details in the figure legend

Line 314: It is not clear what the authors did here. If they collected many soil cores per sites and composited them into three replicate samples per site, that is OK, but they need to specify how the three areas where cores were collected and composited separate from each other. If only 3 soil cores were collected and composited in one sample per site would be an insufficient sampling strategy, missing to capture the natural spatial variability at each site.

Only 3 soil cores were sampled from each site, and they were analysed as distinct replicates and were not composited. The confusing statement about minimising confounding land use treatment effects with spatial factors in line 344 has been amended to "...3 spatially dispersed soil cores were sampled to capture the natural spatial variability at each site". No. of replicates are now mentioned throughout the MS and in the figure legends.

Line 319: Again, this is not clear, and stems from my comment above. Are those three through field replicates of soil samples obtained from the combination of several soil cores, or are they laboratory triplicates of one homogenized soil sample from three soil cores? The authors need to clarify. The first option is OK, the second is not acceptable.

3 distinct soil cores were analysed separately. Changed to "...were assessed in replicated soil samples from each site" (line 349).

Line 321: Please clarify this correction factor. Of course, they did not correlate in calcareous soils because of the presence of significant amount of soil inorganic carbon (SIC). Was SIC quantified, and subtracted from the total C, or were all analyses conducted on total C?

All analysis was done on total C, SIC was not quantified. We observed a linear relationship between total C and LOI ($R^2=0.93$) with the only deviations being for the calcareous soils. For soils from these sites, total C was predicted based on the LOI-C regression model for the non-calcareous sites. Changed in the methods section (line 349-352).

Line 321-323: Were soil maintained at a standard moisture? Differences in field moisture would overwhelm any other difference as control of soil respiration. It is very important that the authors clarify soil moisture levels during incubation.

Soil moisture was not manipulated. We now mention in the text that soils were at field moisture capacity at the time of the assay (line 354, 356, 370). Our results likely reflect real field differences given optimal moisture. CUE may be controlled by soil moisture levels and further experiments addressing specific resistance and resilience of CUE to moisture perturbation are needed. But this was not the focus in our study.

Line 330: How long after the addition of the labelled DOC? In which conditions were the soil stored for this incubation?

Incubations for labelled DOC were performed in the same way as unlabelled DOC, for ~ 16 h in the dark at room temperature (21°C). We've rephrased the sentence for clarity (line 364).

Line 332: Were fresh field soil used for this DNA extraction?

All analyses were performed on homogenised soils from single cores that were frozen at -20°C until analyses (line 346). We now mention at the outset that: "aliquots of the homogenized soil were frozen at -20°C until the following functional analyses".

Line 334: Why was this other incubation performed, couldn't the soils from the previously described incubation used for DNA extractions?

We now mention that both extracts were measured on LC-IRMS (line 370, 374). The extract without labelled substrates serves as an unlabelled control to give background stable carbon isotope ratios. It was also used to assess the taxonomic diversity of bacteria and fungi.

Line 340: How could this equation be applied if (assuming I understood well) the respiration and DNA extractions were performed on different sets of soils? Methods need to be much better described to be assessed for correctness.

Separate aliquots of homogenised soil were used for each of the functional analysis because the procedures are different for each. For eg. for respiration measurement soils needed to be placed in glass vials with rubber septa for GC-IRMS analysis, whereas soils for DNA extractions needed to be in 96-well plates.

Line 378: soil pH, moisture, total C and clay are very few edaphic properties, which are also often correlated between each other. What I find surprising in this study is the total lack of data relative to plant input and chemistry, which are surely largely affected by the above factors, in particular pH, as well as management and that can well be the primary factors affecting microbial traits. What does total soil C refers to,

% total C (ignoring carbonates?) again see my comment above, what was LOI used for?

We've now added more edaphic properties namely N and C:N ratio. We did not collect detailed plant data (see earlier comments above). Soil C refers to % total C. It correlated strongly with LOI except at one calcareous site, indicating presence of very small amounts of carbonates in all but one soil type under investigation. LOI was only used to validate our soil C data. We've added these details in the methods section (line 349-353).

Line 389: To my understanding, only %C and not soil C stocks were determined and used for the analyses. So, the relation is with "soil C density" not storage

We did quantify soil C stocks using bulk density measures but the stock estimates were not satisfactory. Stock estimates can be affected by confounding spatial variation and simple bulk density corrections can fail when comparing different land use systems. Since we do not measure stocks we aren't directly measuring soil carbon storage but it's potential. Rather than talking about mechanisms of soil carbon storage, we now talk about mechanisms of soil carbon accumulation using SOC concentration as a measure of the sum processes of accumulation. We've now changed the language to focus less on storage and more on mechanisms of SOC accumulation with this specific line at the start of the results section (line 104): "We first tested the broader relationships between microbial ecophysiology and soil properties, using C concentration as a proxy for SOC accumulation, to understand how microbial processes determine the mineralisation, assimilation and accumulation of organic matter inputs which underpin SOC formation and loss". We now use the phrase SOC accumulation throughout the MS for the sake of uniformity.

Reviewer #3

Malik et al. produce a set of results that are intriguing and compelling. They are intriguing because they suggest very different roles for microbes in more acidic vs. more neutral soils in soil carbon cycling. Specifically, that in more neutral soils that aggregate properties of the microbial communities (e.g. CUE, turnover) drive soil C formation rates. Whereas in more acidic soils these same microbial variables appear detached from soil C turnover, with acidity instead limiting decomposition rates, meaning that soil C stocks are primarily controlled by abiotic factors. The results are compelling because they draw together a range of different methods for assessing the soils, from state-of-the-art (e.g. proteomics) to well-established and conventional methods (e.g. soil respiration potentials), and the data from each appear to line up to support the authors' interpretation of the patterns observed.

Given the clever way the findings from the different methods were woven together to form logical and interesting arguments in the text, combined with the intriguing

finding that microbes might act as direct controls on soil C dynamics at more neutral pH and yet be disconnected at more acid pH values, I read the paper favourably.

We thank the reviewer for their affirmation

However, I also had numerous questions that need to be addressed before the work can be fully evaluated. My assessment was likely hindered by the fact the Data S1.xlsx file would only download as a corrupt zip file, because many of my questions related to replication and sampling design.

We apologise for this, we assume something went wrong with the manuscript upload/download system. It was a simple 38KB Excel file. Please refer to Table 1 and S1.

However, such information should not only be in supplementary material but readily available by reading, for example, figure legends. As well as rectifying these omissions, I'd ask the authors to consider the following points:

We have now included more details about the sampling sites, land management history, number of replicates, etc. in the methods section as well as in figure legends. We've also added two new tables with site descriptions one in the main manuscript and the other as Supplementary Info.

1. pH is on a log scale. Do microbes see pH or do they see hydrogen ion concentration? If you repeated your analysis with pH back-transformed to H⁺ ion concentration, would any of your conclusions change. For example, would the threshold value remain as pH 6.2?

This is an interesting point. We have already back-transformed to H⁺ ion concentration for the figure 4-fold change analysis to facilitate a log-scaled fold change comparison with other parameters. We now also tried back transformed H⁺ concentrations for regression analyses, but the results do not change. We stick with pH values in the manuscript.

2. Related to pH, I could not find in the Methods how you calculated pH. Given its prominence in the findings, it seems you really should detail this and clarify, for example, if taken in water or in CaCl₂, which would then alter the meaning of 6.2 for people.

pH was measured in deionised water; now added in the methods section line 352. Also, the threshold value of 6.2 is not a special number, it is only pertinent for these collected soils. We mention this caveat in the discussion: "...across a soil pH threshold of ~ 6.2 for the sites assessed here".

3. Figures 2 and 4 are key data figures for establishing a robust empirical basis for the arguments you weave together. But I could not discern the n: please state this on the figure. You do state elsewhere that you had 56 sites and took 3 replicate samples per site, but in Fig. 2 for example you clearly have many more than 56 blue-

circle symbols, leaving me confused. So, are you treating each replicate from a site as independent?

We've now added the n values in the figure legends. Yes, we treat each replicate as an independent point for the initial empirical relationships. Like the reviewer mentions, this allowed us to account for spatial non-independence. For figure 4, we average the values per site and then calculate the fold change as a ratio of low and high intensity land use. This is now mentioned in the methods section (line 416, 437).

4. Similarly, I do not know the n for your statistical models. These should also be stated. And please clarify how are treating the 3 replicate samples per site – it seems you might average them or better still have a random effect to account for potential spatial non-independence.

More info has been presented in the methods section. We've also clarified how we're treating the three replicates, in the figure legends and the methods section.

5. Your statement on line 102 that you treated SOC concentration as a proxy for soil C storage has huge consequences for the arguments that you build. I would argue that you need to carefully go through your paper and much better caveat your conclusions/ language given this assumption.

We did quantify soil C stocks using bulk density measures but the stock estimates were not satisfactory. Stock estimates can be affected by confounding spatial variation and simple bulk density corrections can fail when comparing different land use systems. Since we do not measure stocks we aren't directly measuring soil carbon storage but it's potential. Rather than talking about mechanisms of soil carbon storage, we now talk about mechanisms of soil carbon accumulation using SOC concentration as a measure of the sum processes of accumulation. Throughout the MS, we've now changed the language to focus less on storage and more on mechanisms of SOC accumulation with this specific line at the start of the results section line 103: "We first tested the broader relationships between microbial ecophysiology and soil properties, using C concentration as a proxy for SOC accumulation, to understand how microbial processes determine the mineralisation, assimilation and accumulation of organic matter inputs which underpin SOC formation and loss." We now use the phrase SOC accumulation throughout the MS for the sake of uniformity.

It appears from your methods that you have a single soil collection, and hence a single snapshot of microbial variables from which you make many inferences about the consequences for much longer-term soil C dynamics. Although your arguments are logical and interesting, they should be much more cautious.

It is true that we performed soil collection at single time point. This was aimed at assessing the legacy of land use on microbial physiology. We now mention

that this was a single time point sampling exercise (line 344) and also focus on SOC accumulation and not long-term storage. We agree that seasonality and climate may shift SOC accumulation intensity, but probably would not alter the strong patterns between pH and other edaphic factors, and SOC accumulation.

For example, you have an observational study. How do we then discern whether it is the microbes affecting soil C, or the soil C driving the microbial community properties?

It's probably a two-way cause and effect, with feedbacks. We are now cautious in implying causality with sentences like "CUE and SOC concentration co-varied, indicating the interdependence of microbial growth efficiency and SOC accumulation". Also, in the discussion we already mention the following point suggesting a two-way feedback: "We suggest that cover crops and other restorative conservation approaches that are aimed at increasing plant matter input into soils would over a period of time promote microbially-mediated reiterative soil organic matter formation and decrease the resource limitation and water stress."

We hypothesised that microbial physiology affects its biomass production and that subsequently determines the microbial residue contribution to soil organic matter formation. Using a confirmatory approach, we tried to fit this path of causality in our structural equation modelling. But, we also tried other random combinations like SOC affecting CUE and biomass, but the data didn't fit this path structure.

Further, your arguments rely on greater SOC formation at >6.2 pH but with greater microbial efficiency and turnover would we also not expect greater microbial-mediated decomposition. So, what does that mean for soil C stocks (which you do not actually measure): will more gain be balanced by more loss?

First, we observed that microbial efficiency and turnover were not coupled. For soils at pH >6.2, increase in CUE is linked to increase in SOC but the relationship was opposite for turnover rate and SOC (we've added some lines to clarify this confusion, line 128-136). Higher efficiency may not directly link with higher microbial mediated decomposition. It means higher amount of resources are channelled into growth and biomass production. So, given similar inputs microbial communities with higher CUE will channel more C into biomass and losses will be smaller.

And what exactly is the mechanism of acid protection of soil C, and if it's so important why does microbial biomass correlate strongly with soil C concentration without any change for the pH threshold?

It's not really acid protection of SOC, but we call it acid retardation of microbial decomposition. There is likely higher amounts of undecomposed plant and microbial organic matter in these acidic soils. To this end, we do observe

slower microbial growth in these soils. We've changed the wording that suggested an explicit abiotic control on C storage at lower pH to now include the microbial role. In acidic wet soils, microbial growth and decomposition rates were lower (line 120-133).

Overall, I like the paper and the approach and findings are interesting and compelling; but the over-extrapolation of the results was a strong detraction for me. You do this throughout the script, and an interpretation better constrained by the data would help the paper a great deal, and not lessen the value/ impact of the work.

We've toned down the interpretations and added caveats. We thank the reviewer for their criticism that has helped us improve the MS greatly.

6. Line 139 – strange that you would even think that taxonomic diversity indices would be more strongly linked than community physiological traits. This part seems a little bit of a straw man. Can you back that expectation up with citations, and maybe also present the counter-arguments that physiological traits should matter more?

It has been shown that taxonomic diversity is linked to land use change and the concurrent soil edaphic factors including soil C and pH (Goss-Souza et al., 2017; Thomson et al., 2015; Tian et al., 2017). However, its impact on microbial physiology has not been sufficiently explored. One could expect better functionality with higher number of taxa. However, there is no consensus in the literature on whether taxonomic or functional indicators are linked strongly to edaphic factors. We simply state these linkages here, it is not our intention in this paper to enter the debate as to why higher or lower diversity should affect soil processing.

7. Line 155 – can you give a citation for why you think taxonomic diversity would drive the trait response?

These references (Martiny et al., 2015; Treseder et al., 2011) have been added

8. Line 160 – this is clearly a key argument but I do not follow the logic. Why does a weak link between CUE and biomass, but a strong link from biomass to soil C concentration, suggest abiotic factors dominate?

This indicates decoupling of relationships between CUE, microbial biomass and carbon storage in acidic soils. The doctrine that efficiency and high biomass are associated with organic matter accumulation is therefore not true in acid soils. Here, acidity retards microbial growth and decomposition resulting in higher organic matter accumulation, and high biomass communities exhibiting low efficiency (presumably due to high cellular maintenance requirements for coping with acid stress). We've made these concept and arguments clearer throughout the revised manuscript (line 122-133, 174-178).

9. You mention a few times that pH is an integrated proxy of conditions. Yet many of your arguments are premised on acidity being a stress. Which is it? Is pH acting directly, or via mechanisms such as control on P availability? Please ensure the inferences are internally consistent.

We know that acidity imposes multiple stresses directly and indirectly on microorganisms – unpicking which is playing a role is extremely challenging and not the focus here. Our revised interpretations on the microbial mechanisms at low pH should address this point.

10. The supplementary figures need more explanation to be helpful. For example, in the regression tree pH 6.2 is not identified as a threshold. And what criteria are used to establish slope failure in the coefficient/ r^2 figure? It would seem that the r^2 and coefficient was still pretty strong for where you placed the threshold.

We have now included more details about the slope failure test in the methods section (line 417-420) and Supporting Information legends. We've also altered some of the parameters (eg. the pH window is much narrower now) to get a more robust analytical outcome. We now have a modified figure S2.

Regression tree was not used to identify threshold but to disentangle interactions between CUE and edaphic factors and to identify the best partitioning parameter. Once pH was identified as the best partitioning parameter, slope failure test was used to estimate the threshold value where the CUE-SOC relationship breaks down. Although the relationship is still significant below pH 6.2 (if we associate significance to p-value of <0.05), the R^2 drops dramatically below pH 6.2 (figure S2) which was used as the threshold value.

11. The land use intensification subset is intriguing but given the absence of data on n, it was not clear how this subset really related to the 56 full sites, how many observations you used when you got down to 21 paired locations, and so on. Without such information, your whole design remains somewhat of a mystery.

We've now added more details in the methods section about how the land use comparison was done and how figure 4 was obtained (line 434-442) including a new table in the main text. We apologise for the inaccessibility of the supporting data file with information on the land management history of the sites used for comparison. Our guess is that something went wrong with the file download system. Please refer to the new tables for site descriptions.

References cited in this document:

Goss-Souza, D., Mendes, L.W., Borges, C.D., Baretta, D., Tsai, S.M., Rodrigues, J.L.M., 2017. Soil microbial community dynamics and assembly under long-term land use change. *FEMS Microbiology Ecology* 93, 1–13.

doi:10.1093/femsec/fix109

- Malik, A.A., Dannert, H., Griffiths, R.I., Thomson, B.C., Gleixner, G., 2015. Rhizosphere bacterial carbon turnover is higher in nucleic acids than membrane lipids: implications for understanding soil carbon cycling. *Frontiers in Microbiology* 6, 268. doi:10.3389/fmicb.2015.00268
- Malik, A.A., Roth, V.-N., Hébert, M., Tremblay, L., Dittmar, T., Gleixner, G., 2016. Linking molecular size, composition and carbon turnover of extractable soil microbial compounds. *Soil Biology and Biochemistry* 100, 66–73. doi:http://dx.doi.org/10.1016/j.soilbio.2016.05.019
- Malik, A.A., Thomson, B.C., Whiteley, A.S., Bailey, M., Griffiths, R.I., 2017. Bacterial Physiological Adaptations to Contrasting Edaphic Conditions Identified Using Landscape Scale Metagenomics. *MBio* 8, e00799-17. doi:10.1128/mBio.00799-17
- Martiny, J.B.H., Jones, S.E., Lennon, J.T., Martiny, A.C., 2015. Microbiomes in light of traits: A phylogenetic perspective. *Science* 350.
- Thomson, B.C., Tisserant, E., Plassart, P., Uroz, S., Griffiths, R.I., Hannula, S.E., Buée, M., Mougél, C., Ranjard, L., Van Veen, J.A., Martin, F., Bailey, M.J., Lemanceau, P., 2015. Soil conditions and land use intensification effects on soil microbial communities across a range of European field sites. *Soil Biology and Biochemistry* 88, 403–413. doi:10.1016/j.soilbio.2015.06.012
- Tian, Q., Taniguchi, T., Shi, W.-Y., Li, G., Yamanaka, N., Du, S., 2017. Land-use types and soil chemical properties influence soil microbial communities in the semiarid Loess Plateau region in China. *Scientific Reports* 7, 45289. doi:10.1038/srep45289
- Treseder, K.K., Kivlin, S.N., Hawkes, C. V., 2011. Evolutionary trade-offs among decomposers determine responses to nitrogen enrichment. *Ecology Letters* 14, 933–938. doi:10.1111/j.1461-0248.2011.01650.x
- Zhang, Z., Qu, Y., Li, S., Feng, K., Wang, S., Cai, W., Liang, Y., Li, H., Xu, M., Yin, H., Deng, Y., 2017. Soil bacterial quantification approaches coupling with relative abundances reflecting the changes of taxa. *Scientific Reports* 7, 1–11. doi:10.1038/s41598-017-05260-w

REVIEWERS' COMMENTS:

Reviewer #1 (Remarks to the Author):

I agree with the authors that there is no perfect method to determine microbial biomass C. However, it needs to be stated in the manuscript that C-DNA is used as a proxy for microbial biomass C, which underestimates microbial biomass C, as C-DNA is only a fraction of microbial biomass C (or of CFE-derived microbial biomass C). Using C-DNA as biomass proxy leads to an underestimation of the absolute value of microbial CUE. In the present study the values of CUE are very low, most of them below 0.1. Therefore, at least, the authors need to mention that CUE in the present study are lower than the true values and that they are lower compared to similar approaches that use different microbial biomass estimates (e.g., PLFA or CFE). If the authors have data on microbial biomass C (e.g., PLFA or CFE) available, I suggest that the data could be presented in the supplement and could be briefly discussed despite their limitations.

A remark on the comment of reviewer 2 and the response of the authors regarding the addition of the same ¹³C-labelled substrate to all soils: Recently a new approach became available to measure microbial CUE in soils by the incorporation of ¹⁸O into DNA from ¹⁸O-H₂O (Spohn et al. 2016, cited by the authors). In this approach, microbial CUE estimates are based on the in-situ C concentration and composition and therefore may give "true" CUE estimates, avoiding a bias through substrate preferences/adaptations of the soil microbial communities.

L68: I suggest to replace "hypothesise" by "argue".

Reviewer #3 (Remarks to the Author):

As in the original version of this submission, I very much appreciated the clever way the findings from the different methods were woven together to form logical and interesting arguments in the text, combined with the intriguing finding that microbes might act as direct controls on soil C dynamics at more neutral pH and yet be disconnected at more acid pH values.

In the original submission, I found a good deal of the methodological details lacking, that inference was too causal, and that the implications for carbon cycling were oversold. I note that Reviewer 2 also had a number of critiques. The authors were able to make few changes to address the concerns of Reviewer 2 because they did not have the data on plant communities that the Reviewer was asking for. However, I do not think that this is a problem - Malik et al. make it explicit that they are looking for long-term effects of land use mediated via changes in soil properties. If there were no effects of soil properties, one might ask them to look at plant variables and equally plant variables might add more explanatory power; but to suggest their results are not valid with the absence of plant data is similar to arguing that unless we measure everything in every study then the results are not useful. All results are provisional and build knowledge, and clearly Malik et al. used multiple approaches, measured a subset of soil properties but ones believed to exert strong control, and found strong effects. Further, contrary to the Reviewer's assertion, soil properties such as clay do change with land use intensity (e.g. through erosion, soil mixing, etc - all of which changes the depth at which particular horizons are encountered or can mix them). That is, texture for a particular depth-increment of soil is not insensitive to management. Lastly, contrary to the Reviewer's critique, if the authors had not included clay it would have been an error in design - in multiple regression approaches much of the power arises because you want to know the effect of X on Y, when you have accounted in your model for the effects of A, B and C also on Y. I think an overemphasis on AIC in ecology has begun to blind people to such basic statistical premises, and I suggest the Reviewer pick

up any econometric paper where they commonly have large, observational datasets and multiple predictors to look at their analytical approach.

As such, in addressing my concerns and those of Reviewer 2 (and 1), I think the authors have made a thorough revision and the result is a paper that remains clever in design, important and novel from a findings perspective, and which is now much more complete and balanced. Thanks to Malik et al. for such a thorough revision and an interesting paper.

Response to reviewer comments

Reviewer #1

I agree with the authors that there is no perfect method to determine microbial biomass C. However, it needs to be stated in the manuscript that C-DNA is used as a proxy for microbial biomass C, which underestimates microbial biomass C, as C-DNA is only a fraction of microbial biomass C (or of CFE-derived microbial biomass C). Using C-DNA as biomass proxy leads to an underestimation of the absolute value of microbial CUE. In the present study the values of CUE are very low, most of them below 0.1. Therefore, at least, the authors need to mention that CUE in the present study are lower than the true values and that they are lower compared to similar approaches that use different microbial biomass estimates (e.g., PLFA or CFE). If the authors have data on microbial biomass C (e.g., PLFA or CFE) available, I suggest that the data could be presented in the supplement and could be briefly discussed despite their limitations.

We've now added this sentence at line 150 in the manuscript: "We note that using DNA-C concentration as biomass proxy could lead to an underestimation of total microbial biomass as well as the absolute value of microbial CUE compared to approaches that employ other biomarkers." It is a great suggestion to compare ¹³C incorporation into different microbial fractions. Unfortunately, we do not have data on PLFA or CFE from this study. But we have previously compared and discussed different biomarkers in the following manuscript:

Malik, A. A., Dannert, H., Griffiths, R. I., Thomson, B. C. & Gleixner, G. Rhizosphere bacterial carbon turnover is higher in nucleic acids than membrane lipids: implications for understanding soil carbon cycling. *Front. Microbiol.* 6, 268 (2015).

A remark on the comment of reviewer 2 and the response of the authors regarding the addition of the same ¹³C-labelled substrate to all soils: Recently a new approach became available to measure microbial CUE in soils by the incorporation of ¹⁸O into DNA from ¹⁸O-H₂O (Spohn et al. 2016, cited by the authors). In this approach, microbial CUE estimates are based on the in-situ C concentration and composition and therefore may give "true" CUE estimates, avoiding a bias through substrate preferences/adaptations of the soil microbial communities.

We thank the reviewer for this suggestion. We are aware of this method. It is technologically a bit more demanding and has certain limitations. The method wasn't available to us but we would consider it in future studies.

L68: I suggest to replace "hypothesise" by "argue".

Amended

Reviewer #3

As in the original version of this submission, I very much appreciated the clever way the findings from the different methods were woven together to form logical and

interesting arguments in the text, combined with the intriguing finding that microbes might act as direct controls on soil C dynamics at more neutral pH and yet be disconnected at more acid pH values.

We thank the reviewer for their affirmation.

In the original submission, I found a good deal of the methodological details lacking, that inference was too causal, and that the implications for carbon cycling were oversold. I note that Reviewer 2 also had a number of critiques. The authors were able to make few changes to address the concerns of Reviewer 2 because they did not have the data on plant communities that the Reviewer was asking for. However, I do not think that this is a problem - Malik et al. make it explicit that they are looking for long-term effects of land use mediated via changes in soil properties. If there were no effects of soil properties, one might ask them to look at plant variables and equally plant variables might add more explanatory power; but to suggest their results are not valid with the absence of plant data is similar to arguing that unless we measure everything in every study then the results are not useful. All results are provisional and build knowledge, and clearly Malik et al. used multiple approaches, measured a subset of soil properties but ones believed to exert strong control, and found strong effects. Further, contrary to the Reviewer's assertion, soil properties such as clay do change with land use intensity (e.g. through erosion, soil mixing, etc - all of which changes the depth at which particular horizons are encountered or can mix them). That is, texture for a particular depth-increment of soil is not insensitive to management. Lastly, contrary to the Reviewer's critique, if the authors had not included clay it would have been an error in design - in multiple regression approaches much of the power arises because you want to know the effect of X on Y, when you have accounted in your model for the effects of A, B and C also on Y. I think an overemphasis on AIC in ecology has begun to blind people to such basic statistical premises, and I suggest the Reviewer pick up any econometric paper where they commonly have large, observational datasets and multiple predictors to look at their analytical approach.

Yes, we agree. The focus of this study was to look at detailed microbial physiology and the absence of plant data does not make the results less valid.

As such, in addressing my concerns and those of Reviewer 2 (and 1), I think the authors have made a thorough revision and the result is a paper that remains clever in design, important and novel from a findings perspective, and which is now much more complete and balanced. Thanks to Malik et al. for such a thorough revision and an interesting paper.

We are delighted that the reviewer found our study and its results interesting. We thank the reviewer once again for their reviewing service and constructive criticism.